# Deep Learning in Medical Image Registration: Magic or Mirage?

**Rohit Jena**[1,4]    **Deeksha Sethi**[1]    **Pratik Chaudhari**[1,2,*]    **James C. Gee**[1,3,4,*]

[1]Computer and Information Science    [2]Electrical and Systems Engineering
[3]Radiology    [4] Penn Image Computing and Science Laboratory
{rjena, deesethi, pratikac}@seas.upenn.edu, gee@upenn.edu

## Abstract

Classical optimization and learning-based methods are the two reigning paradigms in deformable image registration. While optimization-based methods boast generalizability across modalities and robust performance, learning-based methods promise peak performance, incorporating weak supervision and amortized optimization. However, the exact conditions for either paradigm to perform well over the other are shrouded and not explicitly outlined in the existing literature. In this paper, we make an explicit correspondence between the mutual information of the distribution of per-pixel intensity and labels, and the performance of classical registration methods. This strong correlation hints to the fact that architectural designs in learning-based methods is unlikely to affect this correlation, and therefore, the performance of learning-based methods. This hypothesis is thoroughly validated with state-of-the-art classical and learning-based methods. However, learning-based methods with weak supervision can perform high-fidelity intensity and label registration, which is not possible with classical methods. Next, we show that this high-fidelity feature learning does not translate to invariance to domain shift, and learning-based methods are sensitive to such changes in the data distribution. We reassess and recalibrate performance expectations from classical and DLIR methods under access to label supervision, training time, and its generalization capabilities under minor domain shifts.

## 1   Introduction

Deformable Image Registration (DIR) refers to the local, non-linear (hence deformable) alignment of images by estimating a dense displacement field. Many workflows in medical image analysis require images to be in a standard coordinate system for comparison, analysis, and visualization. In neuroimaging, communicating and comparing data between subjects requires the images to lie in a standard coordinate system [48, 96, 89, 32, 81, 85]. This assumption universally does not apply when brain image data are compared across individuals or for the same individual at different time points. Anatomical correspondences between diseased patients and normative brain templates help identify and localize abnormalities like tumors, lesions, or atrophy. Failed or anomalous correspondences impact diagnosis, treatment planning, and disease progression monitoring. DIR is also used to capture and quantify biomechanics and dynamics of different anatomical structures including myocardial motion tracking [74, 73, 7], improved monitoring of airflow and pulmonary function in lung imaging [66, 27, 97], and tracking of organ motion in radiation therapy [45, 14, 68, 78]. Latest breakthrough advances in imaging techniques like fluorescence and light-sheet microscopy [36, 69, 29, 98], *in-situ* hybridization, and multiplexing [65, 102] have led to image registration being imperative in advancing life sciences research. Relevant research includes a brain-wide mesoscale connectome of the mouse brain [67], uncovering behavior of individual neurons in *C.*

---

*Equal advising

38th Conference on Neural Information Processing Systems (NeurIPS 2024).

*elegans* [91], building cellular-level atlases of *C. elegans*, *Drosophila melanogaster*, and the mouse brain [105, 90, 96, 76, 71, 13].

Classical optimization-based and learning-based methods are the two reigning paradigms in DIR. Classical DIR methods are based on solving a variational optimization problem, where a similarity metric is optimized to find the best transformation that aligns the images. Most classical methods are formulated without any particular domain knowledge encoded in the optimization problem, and are therefore general and applicable to a wide range of problems. For instance, the popularly known registration toolkit ANTs [5] has been successfully applied to structural *and* functional neuroimaging data [48, 104, 43], CT lung imaging [66], cardiac motion modeling [53], developmental mouse brain atlases utilizing MRI and light sheet fluorescence microscopy [50] with virtually no change in the optimization algorithm. However, classical iterative methods have slow convergence, their performance is limited by the fidelity of image intensities, and they cannot incorporate learning to leverage a training set containing weak supervision such as anatomical landmarks, label maps or expert annotations. Deep Learning for Image Registration (DLIR) is an interesting paradigm to overcome these challenges. DLIR methods take a pair of images as input to a neural network and outputs a warp field that aligns the images, and their associated anatomical landmarks. The neural network parameters are trained to minimize the alignment loss over image pairs and landmarks in a training set. During inference, an image pair is provided and the network regresses a warp field. A primary benefit of this method is the ability to incorporate weak supervision like anatomical landmarks or expert annotations during training, which performs better landmark alignment without access to landmarks at inference time.

**Motivation**   However, the benefits of using DLIR methods over classical DIR methods in terms of accuracy or robustness to domain shift are still topics with no clear consensus. Several DLIR methods claim that architectural choices and loss function design combined with amortized optimization of neural network parameters significantly outperform classical methods  [63, 61, 17].  On the contrary, classical iterative methods that leverage implicit or explicit conventional priors have shown to outperform most deep learning methods on other challenging datasets [100, 79]. For example, in the context of lung registration, an implicit neural optimization method surpasses every deep learning baseline on the DIR-lab dataset [100]. In EMPIRE10 challenge without access to labeled data [66], classical methods are highly performant compared to deep learning methods.  In the ANHIR histology registration challenge [12], the best performing algorithms were classical methods, and the deep learning method was fast and performed well, but did not have good generalization capabilities. Mok *et al.* [62] also mention that deep baselines typically fail 'spectacularly' on out of distribution data, and classical methods like Elastix and ANTs come out on top. However, these observations are relatively unstructured and not studied directly. The confounding variable of using labelmap supervision has urged the Learn2Reg 2024 LUMIR challenge [35] to be performed on fully unsupervised data. In our own empirical evaluations, we found that classical methods typically outperform deep methods under certain conditions and assumptions. Image registration is *NP-hard* being a non-convex optimization problem, and approximating the solution of NP-hard problems with deep learning methods is not guaranteed to be optimal, or even a minima of the registration loss at test-time.  Deep learning methods also claim to provide amortized optimization since classical methods are extremely slow to run, however, modern GPU implementations [55, 59, 41] have patched this shortcoming of classical methods while providing state-of-the-art performance.

**Contributions.**   The conditions needed for either paradigm to perform well over the other are clouded and not explicitly outlined in the existing literature. This has prolonged the tug-of-war between classical and deep learning methods. We perform a more structured problem setup and empirical evaluation to determine consensus on the benefits and limitations of each paradigm. First, we observe a strong correlation between the mutual information between per-pixel intensity and label maps, and the performance of classical registration methods. This strong correlation hints to the fact that the Jacobian projection in DLIR methods is unlikely to affect this correlation, and therefore, the performance of DLIR methods in the unsupervised setting. We empirically verify this hypothesis on a variety of state-of-the-art classical and DLIR methods, and address instrumentation bias in the existing literature. Secondly, since the label map is a deterministic function of the intensity image, DLIR methods can learn to perform better label matching when this constraint is enforced during training, by implicitly discovering the label map within the network features and predicting a warp field that minimizes the alignment error between label maps. This is a key strength of DLIR methods, that classical methods cannot leverage. Third, we show that even though learning methods implicit capture semantic information from the image which is not explicitly captured by classical methods,

this additional feature learning does not translate to invariance to domain shift, and DLIR methods are brittle to these changes. These empirical findings allow us to reassess and recalibrate performance expectations from classical and DLIR methods, using a systematic, unbiased and fair evaluation.

## 2 Related Work

### 2.1 Classical Optimization-Based Methods

Classical image registration algorithms employ iterative optimization on a variational objective to estimate the dense displacement field between two images. Some of the earliest approaches to deformable registration considered models for small deformations using elastic deformation assumptions [51, 23, 8, 31, 30, 20, 21], conceptualizing the moving image volume as an elastic continuum that undergoes deformation to align with the appearance of the fixed image. This was in conjunction with alternate formulations based on fluid-dynamical Navier-Stokes [22, 21] and Euler-Lagrange equations [2, 11, 4, 56, 58] and their subsequent optimization strategies. The seminal work of Beg.*et al.* [11] introduces an explicit Euler-Langrange formulation and a metric distance on the images as measured by the geodesic shortest paths in the space of diffeomorphisms used to transform the moving image to the fixed image. However, storing the explicit velocity fields is expensive in terms of compute and memory. This limitation motivated semi-Langrangian formulations [4, 3] to avoid storing velocity fields explicitly, and only storing the final diffeomorphism. ANTs [5, 1] is a widely used toolkit that employs the Euler-Langrange formulation with a symmetric objective function [2]. Yet another approach is to interpret deformable registration as an optical flow problem [70, 103], leading to the famous Demons algorithm and its diffeomorphic and symmetric variants [106, 93, 92, 95] implemented as part of the Insight Toolkit (ITK) [40, 25]. However, most of these methods are still computationally expensive to run owing to their CPU implementations. Recently, modern implementations leverage the massively parallelizable nature of the registration problem to run on GPUs, leading to orders of magnitude of speedups while retaining the robustness and accuracy of the classical methods [55, 59, 41]. However, as we show in Section 4, the registration performance of classical methods is limited by the fidelity of image intensities.

### 2.2 Deep Learning for Image Registration

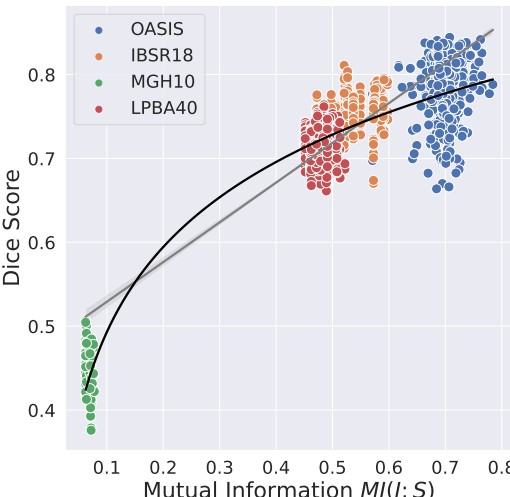

Figure 1: **Correlation between Dice Score and Mutual Information.** Classical registration methods like ANTs show a strong correlation between the Dice Score of registered pairs, and the mutual information between the corresponding image and label across 4 brain datasets.

In contrast to most classical methods, earliest Deep Learning for Image Registration (DLIR) methods employed supervised learning for registration tasks [15, 49, 77, 80] where the deformation field is obtained either manually or from a classical method. Voxelmorph [9] was one of the first approaches that introduced unsupervised learning for registration of in-vivo brain MRI images. Subsequent research expanded upon this paradigm, exploring diverse architectural designs [18, 52, 42, 62], loss functions [109, 108, 44, 24, 60, 107, 75, 16], and formulations based on incorporating inverse-consistency or symmetric transforms [61, 46, 47, 83, 109]. However, hyperparameter tuning became a challenge for DLIR methods since the methods had to be retrained for every new value of the regularization parameter. This motivated techniques such as conditional hyperparameter injection which addressed hyperparameter tuning [64, 38], while domain randomization and fine-tuning [37, 88, 72, 28] aimed to addressed generalizability of DLIR methods across domains. Recently, pretrained or foundation models are also proposed to address the generalizability of DLIR methods across different imaging and anatomy [54, 84]. However, these methods perform a monolithic prediction of the warp field from the input images, losing feedback from the intermediate stages of the registration process as done in classical methods. To refine the warp fields, recurrent or cascade-based archi-

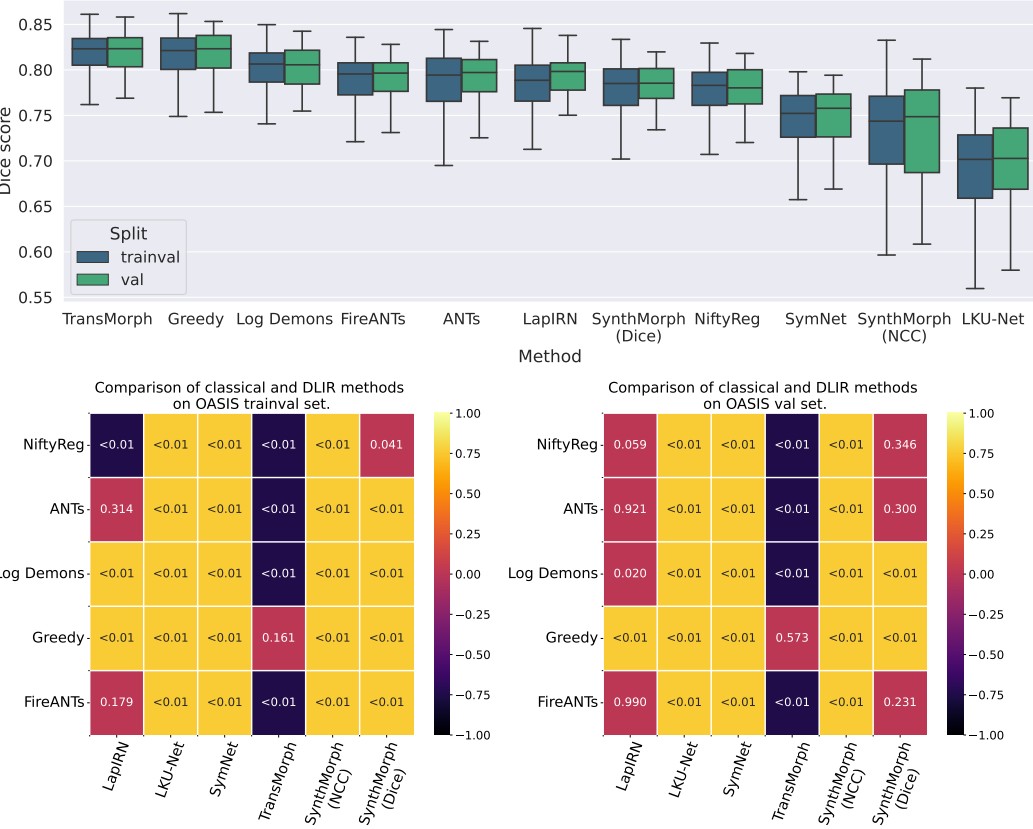

Figure 2: **Performance of classical and unsupervised DLIR methods on OASIS data.** Boxplots (**top**) show that classical methods on average are ranked higher than DLIR methods, both on the *trainval* and *val* splits. Interestingly, the performance of unsupervised DLIR methods does not improve on the *trainval* split compared to *val* split – showing that deep learning does not have an intrinsic advantage in label alignment. Tables (**bottom**) of p-values show the results of a pairwise two-sided t-test between the performance of classical and DLIR methods on the *trainval* and *val* splits. ▉ denotes a cell where the classical method is significantly better than the DLIR method ($p < 0.01$), a ▉ denotes the opposite, ▉ denotes no significant difference. Most of the cells are ▉, indicating that classical methods are significantly better than DLIR methods.

tectures were proposed [108, 109, 107, 16]. However, cascade-based methods create a substantial memory overhead due to backpropagation through cascades and storage of intermediate volumes [6]. Another promising avenue is to leverage deep implicit priors [87] within optimization frameworks to improve the performance of optimization methods or incorporate implicit constraints of the optimized warp field [101, 99, 44, 39]. We refer the reader to [26, 33] for a comprehensive review of image registration techniques.

Despite the plethora of architectural formulations, loss functions, and output representations proposed in Deep Learning for Image Registration methods, we identify that these methods are highly sensitive to the domain gap between the distributions of training and test data, and in the unsupervised case, do not provide any benefit in terms of performance over classical methods. Their primary benefit is their ability to incorporate weak supervision like anatomical landmarks or expert annotations during training, which performs better landmark alignment on unseen image pairs (from the same distribution) without access to landmarks at inference time.

## 3  Preliminaries

We rehash the image registration problem statement to unify both classical and deep learning methods. Consider a dataset of image pairs $\mathcal{D} = \{(I_f^{(n)}, I_m^{(n)}) \mid n \in \mathbb{N}, 1 \leq n \leq N \}$, where $I_f^{(n)}$ and $I_m^{(n)}$ are the fixed and moving images defined over a spatial domain $\Omega \in \mathbb{R}^d$. We drop the superscript

$n$ for simplicity. Also consider segmentation maps $S_f$ and $S_m$ for the fixed and moving images, respectively, defined over $\Omega$. Given a family of transformations $T(\Omega)$, the goal of image registration is to estimate transformations $\varphi_\theta(f, m) \in T(\Omega)$ parameterized by $\theta$ that minimize the following objective:

$$\arg\min_\theta \sum_{f,m} \mathcal{L}(I_f, I_m \circ \varphi_\theta(f, m)) + \mathcal{R}(\varphi_\theta(f, m)) \tag{1}$$

where $\mathcal{L}$ is a dissimilarity function such as mean squared error, or negative local cross correlation, and $\mathcal{R}$ is a regularization term that encourages desirable properties of the transformation, such as smoothness or elasticity. We call Eq. (1) the *image matching* objective, since the transformations only need to align the intensity images. We can also call this the *unsupervised* objective, since it does not require any labeled data. If a suitably chosen label alignment loss $\mathcal{D}$ is added as well, the optimization problem becomes:

$$\arg\min_\theta \sum_{f,m} \mathcal{L}(I_f, I_m \circ \varphi_\theta(f, m)) + \mathcal{D}(S_f, S_m \circ \varphi_\theta(f, m)) + \mathcal{R}(\varphi_\theta(f, m)) \tag{2}$$

We call Eq. (2) the *label matching* objective, or a *weakly-supervised* objective. The image matching objective can subsume both DLIR and classical methods by choosing

$$\varphi_\theta(f, m) = \begin{cases} f_\theta(I_f, I_m), & \text{for deep networks,} \\ \varphi_{(f,m)}, & \text{for classical methods.} \end{cases} \tag{3}$$

where $f_\theta$ is a deep network parameterized by $\theta$ and $\varphi_{(f,m)}$ are optimizable free parameters that are indexed by the 2-tuple $(f, m)$, i.e. $\theta = \bigcup_{f,m}\{\varphi_{(f,m)}\}$. In this paper, we consider methods that solve Eq. (1) using gradient-based methods. The gradient of Eq. (1) with respect to $\theta$ is given by (we remove the $\mathcal{R}$ term for simplicity):

$$\frac{\partial \mathcal{L}}{\partial \theta} = \sum_{f,m} \frac{\partial \mathcal{L}}{\partial \varphi_\theta(f, m)} \frac{\partial \varphi_\theta(f, m)}{\partial \theta} \tag{4}$$

The first term $\frac{\partial \mathcal{L}}{\partial \varphi_\theta(f,m)}$ is the training signal from the dissimilarity function which does not depend on the parameters $\theta$ for a given value of $\varphi_\theta(f, m)$ and choice of $\mathcal{L}$. The second term $\frac{\partial \varphi_\theta(f,m)}{\partial \theta}$ is the Jacobian of the transformation with respect to the parameters, which is a projection of the gradient from the space of warp fields to the space of arbitrary parameters. For classical methods, the Jacobian is the identity matrix, for deep networks it is determined by the functional relationship of the output with respect to network parameters. Therefore, the difference in training dynamics and overall performance gap between classical and deep learning methods is likely to be attributed to the choice of $\frac{\partial \varphi_\theta(f,m)}{\partial \theta}$.

## 4 Unsupervised DLIR does not improve label matching performance

A speculated claim of deep learning methods is that they can provide better label matching performance by simply training a network to minimize Eq. (1) in an unsupervised setting. Such improvements are claimed to come from architectural designs, which correspond to choice of Jacobian $\frac{\partial \varphi_\theta(f,m)}{\partial \theta}$. A variety of architectures and parameterizations [17, 63, 64, 62, 34, 82, 101] have been proposed to this effect. **However, we show that this is not the case.**

Image matching objectives ensure that intensities from the moving image are displaced to locations in the fixed image where they are most similar, without regard for alignment for any higher order structures. Intuitively, this will ensure label matching only to the extent that the intensity is predictive of the label. If an intensity value strongly corresponds to a particular label, then image matching will lead to label matching. Similarly, if a given intensity value corresponds to multiple possible labels, then image matching does not tell us which labels are matched via the image matching objective. More formally, considering the per-pixel intensity $i$ and labels $s$ as random variables, one can compute the mutual information between the intensity and label maps, denoted as $MI(i; s)$ to determine the predictability of one from the other. We now show that the label matching performance of classical methods is highly correlated with $MI(i; s)$. We consider a widely used classical method, ANTs [2, 5], to eliminate the effect of any Jacobian term. We consider four brain datasets - OASIS, LPBA40, MGH10, and IBSR18, which are acquired under different scanners, under different resolutions, and have different preprocessing, labelling and postprocessing protocols [57, 48]. For

each dataset, we use ANTs for registering all pairs within the dataset and then evaluate the Dice score as an indicator of label matching performance. For each image $I$ and its corresponding label map $S$, we compute the probability maps $p(i), p(s), p(i, s)$ using histogram binning, followed by the mutual information $MI(i; s) = H(s) - H(s|i)$. A Pearson's correlation coefficient between the Dice scores and the mutual information of the image and label (Fig. 1) reveals a strong linear ($\mathbf{r = 0.886}$) and logarithmic ($\mathbf{r = 0.933}$) relationship between the two quantities, shown by the gray and black lines respectively. Image matching improves label matching performance *only to the extent of the information about the label obtained from the image* (i.e. $MI(i; s)$). At a first glance, the Jacobian term $\frac{\partial \varphi_\theta(f, m)}{\partial \theta}$ seemingly does not have a role in improving this mutual information further.

**Empirical Validation.** We verify this claim empirically on the OASIS dataset, by minimizing Eq. (1) in both DLIR and classical methods. We split the OASIS dataset into a training set of 364 images and a validation set of 50 images. We choose 50 instead of 20 images as in the original split [35] to compute statistical significance. Dice score over 35 subcortical structures is used as the label matching metric. We choose SynthMorph [37], LapIRN [63], SymNet [61], LKU-Net [42] and TransMorph [19] as state-of-the-art DLIR baselines and ANTs [5], NiftyReg [59], Symmetric Log Demons [94], Greedy [106], FireANTs [41] as state-of-the-art classical baselines. For all DLIR methods, we use pretrained models if they are trained with Eq. (1), or train them with the architecture and hyperparameters provided in their original source code. The only exception is SynthMorph, which is trained on synthetically generated data and Dice loss of its corresponding synthetic labels (`shapes-sm` model). To compare SynthMorph's domain generalization capabilities with only the image matching objective, we add another model, dubbed '`shapes-sm-ncc`' that is trained on synthetically generated data as in the original pretrained model, but with the normalized cross-correlation of the aligned synthetic images. For all classical methods, we follow their recommended hyperparameters and run till convergence. All experiments are run on a cluster with 2 AMD EPYC 7713 CPUs and 8 NVIDIA A6000 GPUs.

**Results.** For all methods, we compute the Dice score of all 35 subcortical regions on images in the validation set (denoted as *val*), and all images (denoted as *trainval*). These Dice scores are sorted by median validation performance in Fig. 2(top). Moreover, we perform a two-sided t-test for each *(classical, DLIR)* pair, both on the trainval and validation sets, shown in Fig. 2(bottom). Fig. 2 shows the following conclusions: (a) the top performing classical method (Greedy) and the top performing DLIR method (TransMorph) achieve similar label matching performance on the val and the trainval set, i.e. the differences are *not* statistically significant ($p = 0.161$), (b) classical methods almost always perform better than DLIR methods, even on the training set showing that the Jacobian term does not improve label matching more than the mutual information between the image and label, and (c) for unsupervised DLIR methods, there is no improvement label matching performance in the training set compared to val set. The only role of the Jacobian term is to perform amortized learning, but without supervised objectives, this does not guarantee any additional boost in label matching.

**The effect of instrumentation bias.** The astute reader may observe that this result is in contrast to results shown in prior literature [61, 63, 101, 19, 10]. We note that this is due to instrumentation bias [86], where the baselines' performance may be misrepresented due to changes in hyperparameters, early stopping, or different preprocessing protocols. For instance, [10] mention that the default parameters of ANTs are not optimal, and choose a very different set of parameters (a Gaussian smoothing of 9 pixels, followed by an extremely small 0.4 pixels at the next scale). By stark contrast, we found the recommended parameters to work extremely well for all datasets considered in this paper. We speculate that these changes are done to tradeoff accuracy for speed, since classical methods converge slowly. However, this leads to misrepresentation of the performance of classical baselines. We found much better results (Fig. 2) for classical baselines simply by using their recommended scripts. We compare the discrepancy in performance between the baselines reported in the literature and the ones we obtained in Fig. 3. We follow the guidelines in [86] to evaluate all methods. To ensure our work does not introduce its own instrumentation bias for DLIR baselines, we compare the performance of our trained/pretrained models to the ones reported in the literature (Fig. 3). We make all evaluation scripts and trained models public[2] to encourage fairness and transparency in evaluations.

---

[2]https://github.com/rohitrango/Magic-or-Mirage/

| Evaluation of classical methods reported by baselines | | | | | |
|---|---|---|---|---|---|
| **Method** | **Evaluated Baseline** | **Statistic** | **Reported value** | **Our eval** | **Difference** |
| SymNet | ANTs | Mean | 0.680 | 0.787 | 0.107 |
| PIRATE | ANTs | Mean | 0.699 | 0.787 | 0.088 |
| LapIRN | Demons | Mean | 0.715 | 0.802 | 0.087 |
| LapIRN | ANTs | Mean | 0.723 | 0.787 | 0.064 |
| NODEO | Demons | Mean | 0.764 | 0.802 | 0.038 |
| NODEO | ANTs | Mean | 0.729 | 0.787 | 0.058 |
| Voxelmorph | ANTs | Mean | 0.749 | 0.787 | 0.038 |
| Voxelmorph | NiftyReg | Mean | 0.755 | 0.776 | 0.021 |
| SynthMorph | ANTs | Median | 0.770 | 0.797 | 0.027 |
| Evaluation of DLIR baselines reported by us | | | | | |
| **Method** | **Dice supervision** | **Statistic** | **Reported value** | **Our eval** | **Difference** |
| SynthMorph | - | Median | 0.780 | 0.785 | 0.005 |
| TransMorph-Regular | ✓ | Mean | 0.858 | 0.855 | -0.003 |
| LKU-Net | ✓ | Mean | 0.886 | 0.904 | 0.018 |
| LapIRN | ✗ | Mean | 0.808 | 0.788 | -0.020 |
| SymNet | ✗ | Mean | 0.743 | 0.748 | 0.005 |

Figure 3: **Instrumentation bias in evaluation of image registration algorithms.** We highlight a significant difference in evaluation metrics reported by baselines and our evaluation on the OASIS validation dataset. This difference can be attributed to deviation in hyperparameters from the recommended parameters or early stopping to save time. In either case, this misrepresentation leads to incorrect conclusions about the performance of the algorithm. The reported dice scores are anywhere from 2 to 10 Dice points lower than our evaluation, showing a non-trivial instrumentation bias. We report our own evaluation of DLIR algorithms and compare them with reported values to avoid introducing instrumentation bias in our evaluation.

## 5 Supervised DLIR methods demonstrate enhanced label matching

When label matching is introduced as an objective in Eq. (2), DLIR methods show superior performance than classical methods. Unlike the previous discussion, where only a pixelwise definition of $MI(i; s)$ was used to quantify the coaction of image intensities and label maps, we consider the entire image $I$ and label volume $S$ as high-dimensional random variables. Label maps are now a deterministic function of the image, i.e. $S = f(I)$, where $f$ is the labelling protocol. In addition to image intensity, label maps are a function of morphological features, location, contrast, and the labelling protocol itself. When trained with the label maps as extra supervision, the network can infer these deterministic relationships to output a warp field that maximize both image similarity and label overlap. Classical intensity-based methods, on the other hand, do not have any mechanism to encode this additional relationship. Aligning intensities or intensity patches discards any functional relationship between high-level image features and labels. To show this, we repeat the same experiment setup as in Section 4 on the same splits, but with the label matching objective added as well.

**Results.** Fig. 4(top) shows the Dice scores for supervised classical and DLIR methods trained on the OASIS dataset, sorted by median validation performance. In this case, state-of-the-art DLIR methods outperform classical methods by a large margin, with notably higher Dice score on the *trainval* set than the *val* set, due to overfitting to the label matching for the training set. This is unlike unsupervised DLIR, where there was no improvement in label matching performance on the training set, emphasizing the fact that performing amortized training does not improve label matching performance by itself. These differences are statistically significant, with the exception of SymNet, which diverged under many training settings with the Dice loss, and only works marginally better than its unsupervised counterpart. SynthMorph is not trained on real data, and is added only as a reference for domain-agnostic performance.

This is an unsurprising result – the label matching objective provides additional training signal to the registration task, which is a highly ill-posed problem. Classical methods cannot incorporate this additional signal from a training dataset, and learning-based methods exploit this to achieve better registration on unseen data. Classical methods are, however, agnostic to modalties, intensity distributions, voxel resolutions, and anisotropy. The same registration algorithm (with possibly modified parameters) is applied to datasets with different characteristics, and they still retain their state-of-the-art performance. A related question arises for DLIR methods trained with label matching – does label matching performance transfer to other datasets?

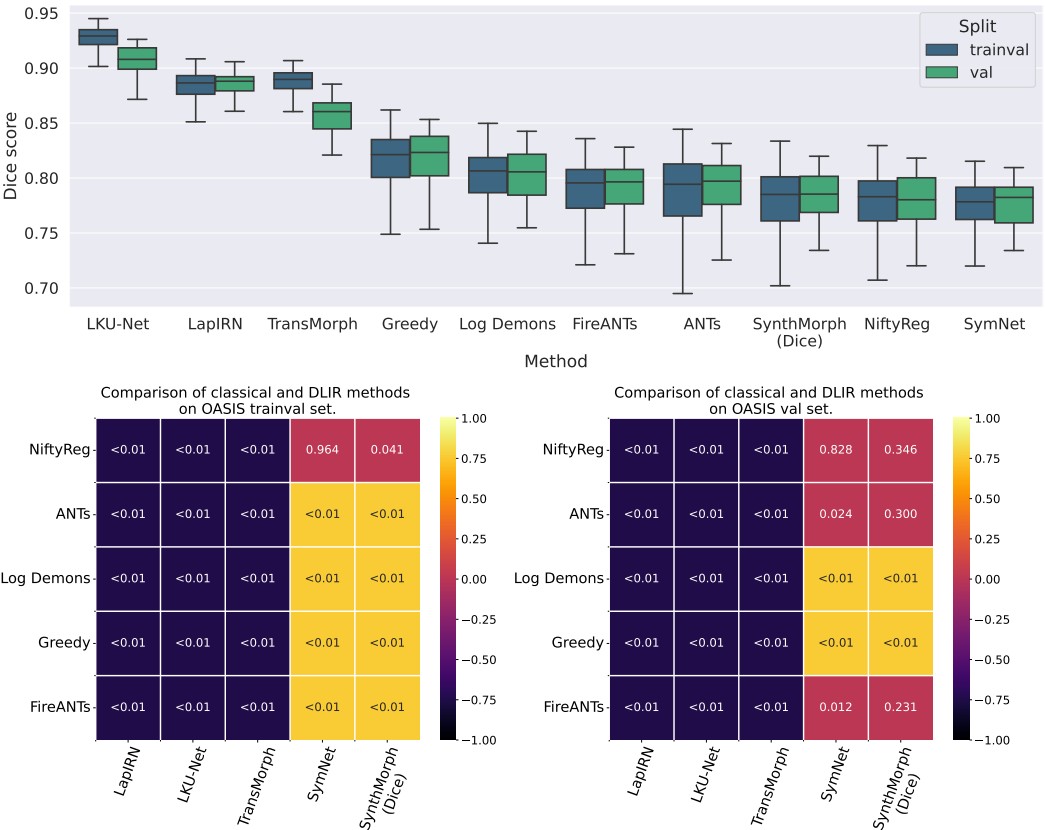

Figure 4: **Performance of classical and supervised DLIR methods on OASIS data.** Boxplots (**top**) show that DLIR methods show superior performance compared to classical methods. Unlike the unsupervised case, the effect of overfitting is clearly visible in the gap between the *trainval* and *val* splits. Tables (**bottom**) of p-values show the results of a pairwise two-sided t-test between the performance of classical and DLIR methods on the *trainval* and *val* splits. ▨ denotes a cell where the classical method is significantly better than the DLIR method ($p < 0.01$), a ▨ denotes the opposite, ▨ denotes no significant difference. State-of-the-art DLIR methods show significantly better performance than classical methods when label supervision is added.

## 6 DLIR methods do not generalize across datasets

A key strength of classical optimization registration algorithms is their agnostic nature to the image modality, physical resolution, voxel sizes, and preprocessing protocols. Most DLIR methods, on the contrary, have been evaluated extensively on the same distribution of validation datasets as the training data, it is unclear if the performance improvements transfer to other datasets of the same anatomy. To this end, we evaluate the performance of both the classical and DLIR methods on four brain datasets – CUMC12, LPBA40, MGH10, and IBSR18. These datasets represent community-standard brain mapping challenge data [48] for a comprehensive evaluation of 14 nonlinear classical registration methods, across various acquisition, preprocessing and labelling protocols. For all datasets, we follow the preprocessing steps followed by [48].

Each dataset contains a different set of labeled regions acquired manually using different labeling protocols. For each dataset, all previously considered registration algorithms are run on all image pairs, and the mean Dice score over all labeled regions is computed. The methods are then sorted by median validation performance in Fig. 5. For DLIR methods, we plot the performance with models trained with and without the label matching loss in the OASIS dataset, shown as blue and green boxplots respectively. Across all datasets, FireANTs, Greedy, ANTs and NiftyReg consistently perform better than DLIR methods. Among the DLIR methods, SynthMorph performs consistently better due to its domain-agnostic training paradigm. Remarkably, even though DLIR methods outperform classical methods on the OASIS dataset with label matching objective, the performance does not transfer to other datasets, even compared to its own unsupervised variant. This is a negative

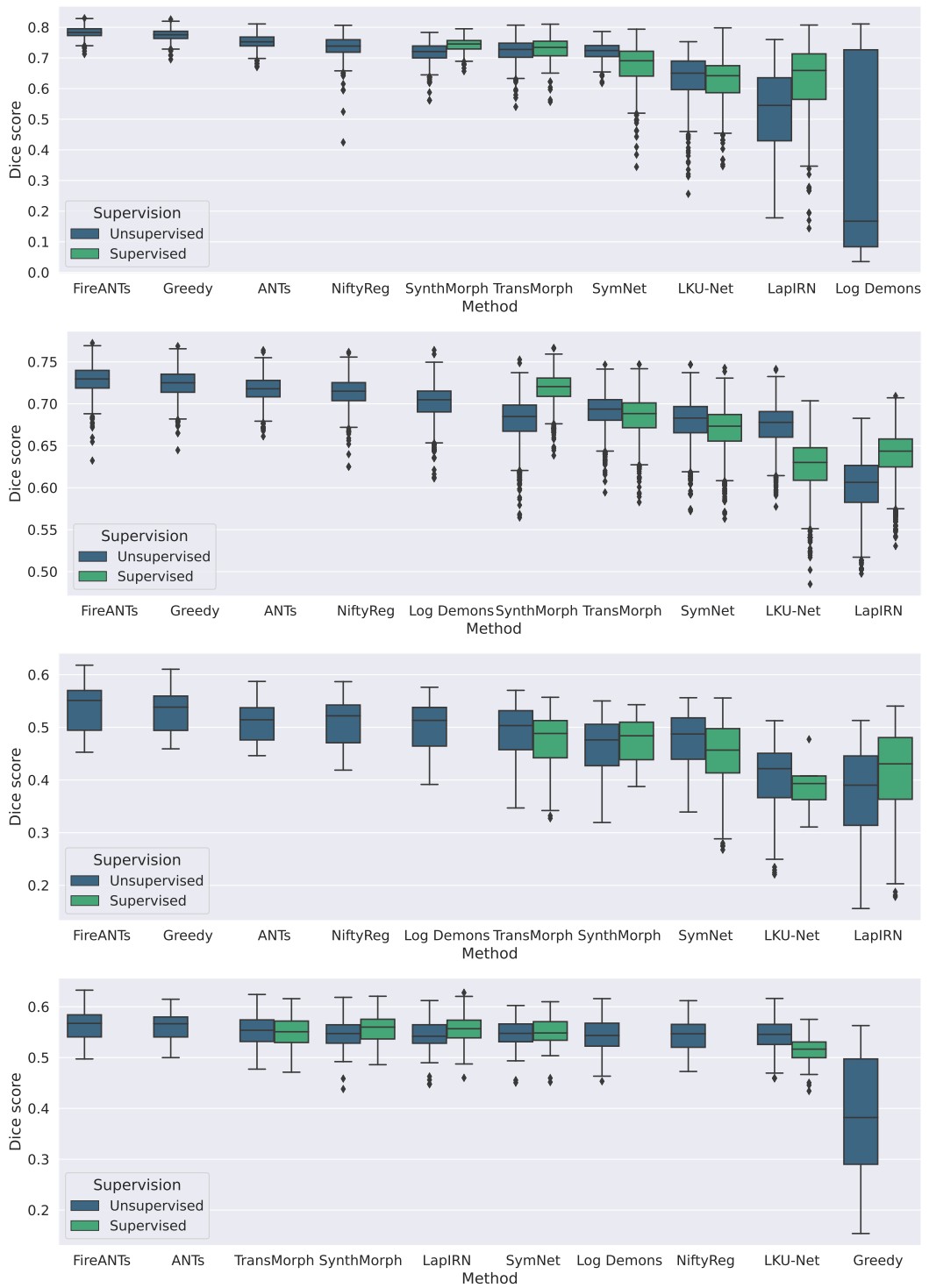

Figure 5: **Classical methods retain robustness across different datasets.** Boxplots show the performance of classical and DLIR methods trained on the OASIS dataset, on four T1-brain datasets. For DLIR methods, we plot the performance of the supervised and unsupervised models. Across all datasets, FireANTs and ANTs consistently outperform DLIR methods, showing robustness to domain shift. Among DLIR methods, SynthMorph and TransMorph show robust performance, and training with label matching objective does not lead to significant improvement.

result – implying that to improve performance on a new dataset, one must collect label maps from that dataset and retrain the model – existing collections of label maps are not sufficient to improve performance on new datasets. Unlike in tasks like segmentation, deep methods do not transfer their performance to out-of-distribution datasets, even with the same resolution, and the expected performance hierarchy does not hold

- **Expected**: Supervised DLIR ID > Supervised DLIR OOD > Classical
- **Observed**: Supervised DLIR ID > Classical > Supervised DLIR OOD

Practitioners should therefore be cautious when using prediction-based DLIR methods, especially when the training data is not representative of the test data, regardless of the presence of label maps.

## 7 Discussion

This study aims to provide a systematic and unbiased investigation of the performance of classical and DLIR methods under access to label supervision, and their generalization capabilities under small domain shifts. Preceding experiments show that classical methods provide an unprecedented level of robustness and generalizability across datasets, but are limited by the fidelity of the image matching objective. Supervised DLIR methods provide a promising step towards improving registration performance of anatomical regions by implicitly discovering these structures and predicting appropriate warp fields within the network architecture. However, this anatomical-awareness on the training dataset does not help in generalizing to other datasets, limiting the practical utility of these methods. The usability of anatomical landmarks and labelmaps to obtain domain-invariant registration performance still remains an open research problem. These results also have profound implications for annotated data collection and challenges the notion that large labeled datasets ensure robust generalization.

Although our study is performed on inter-subject registration with in-vivo neuroimaging datasets, none of our analysis, baselines, and evaluation make any domain or subject-specific modeling assumptions, and the datasets being community-standard benchmarks, the results are valuable and general, both within the neuroimaging and the biomedical communities at large. At the current state, a practitioner should choose predictive DLIR methods only if they have access to a large labeled dataset, *and* their application is limited to the same dataset distribution. In all other cases, classical optimization-based methods are the more accurate and reliable choice, even if labeled data exists but is not representative of the test data.

### 7.1 Limitations

Our work performs a comprehensive evaluation of state-of-the-art registration algorithms on a variety of neuroimaging datasets. However, our work does not consider hybrid methods, or representations that use optimal matching criteria based on correlation volumes or sparse correspondence features. Although our work considers large-scale community-standard neuroimaging datasets, the performance of these algorithms may differ on other anatomy or modalities. Our study also considers inter-subject registration only, although no method or evaluation incorporates any subject-specific assumptions. The effects on multimodal registration are not considered in this work. However, our work serves as a foundational step toward a more nuanced discussion on the longstanding technical challenges in image registration, and representations that are effective in mitigating these problems.

## Acknowledgements

This work was supported by the National Institutes of Health (NIH) under grants RF1-MH124605, R01-HL133889, R01-EB031722, U24-NS135568.

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
