# OpenReview forum: "Deep Learning in Medical Image Registration: Magic or Mirage?"
_NeurIPS.cc/2024/Conference — NeurIPS 2024 poster_

### Official Review · Reviewer_Aeaf · 2024-07-10

**Soundness:** 3
**Presentation:** 3
**Contribution:** 3
**Rating:** 5
**Confidence:** 3

**Summary:**

The paper delves into the comparison between classical optimization and learning-based medical image registration methods. Some valuable insights are proposed and the authors propose a general recipe to choose the best paradigm for a given registration problem.

**Strengths:**

1. The motivation behind the work is clear. The observation is in detail.
2. Rich experiments discover the regular and make people rethink the development of learning-based registration methods.
3. The illustrations are clear and easy to understand.
4. Clear conclusions are obtained for registration paradigms in different conditions.

**Weaknesses:**

1. New solutions or deeper insights are missed.
2. Findings in Section 4,5,6 are somewhat valuable, but the relevancy between different phenomena is not very clear. The author failed to point out what should we do or what we may further research for unsupervised or supervised DLIR methods.

**Questions:**

What are the potential directions of future work based on this paper?

---

> ### Author Rebuttal · Authors · 2024-08-06
>
> We thank the reviewer for their positive and helpful feedback, we are glad to find they found the paper having a clear motivation, detailed observations, clear illustrations and easy-to-read paper. We believe we have addressed all the remaining concerns and hope the reviewer increases their score and advocates for the acceptance of our paper.
>
> **”New solutions or deeper insights are missed.”, “Findings in Section 4,5,6 are somewhat valuable, but the relevancy between different phenomena is not very clear”**
>
> Our work serves as a bedrock for contextualizing the performance of current state-of-the-art deep learning and classical registration methods, and identifying the critical confounding variables responsible for the performance gaps observed in DLIR and classical methods. We show empirically the two confounding variables – instrumentation bias of classical methods, and label supervision are majorly responsible for the superior DLIR performance of in-distribution data.
>
> Our work provides an independent, fair and unbiased evaluation of both DLIR and classical methods, and show that in the unsupervised case, classical baselines outperform DLIR methods without having any in-distribution data – making it useful in the low-data regime. Furthermore, since anatomical labels are an intrinsic property of the anatomy (hence being domain-invariant), one may expect label supervision to add some robustness to domain shift. However, this is not the case (Sec 6).
>
> These experiments naturally imply a recipe to use for a given instance of registration (mentioned in Sec 7). Moreover, these results also allow us to reconsider deploying expensive data collection pipelines, especially when domain shift may be expected at test time (as a consequence of the finding in Sec 6). Moreover, observations in Sec.4. and Sec.6. serve as a problem statement for DLIR-based methods to alleviate these limitations. We urge the reviewer to also read more details about this in the common rebuttal.
>
> **What are the potential directions of future work based on this paper?**
>
> We appreciate the reviewer's interest in future research directions. Our limitations section outlines several avenues for further investigation. This study can serve as a foundational basis for examining and disentangling the variables responsible for robust performance in various contexts, such as:
> Multimodal Registration: Extending our analysis to multimodal registration scenarios to explore the applicability of our findings across different imaging modalities.
> Non-Brain Anatomy: Investigating the generalizability of our results to other anatomical regions beyond brain MRI.
> Diverse Registration Settings: As suggested by Reviewer 23gn, conducting experiments on inter-subject, intra-subject, and atlas-based registration to provide a comprehensive understanding of performance across different registration paradigms.
> We have also incorporated these future work suggestions more explicitly in the revised paper to guide readers on how our findings can be extended and applied in broader contexts.

---

> > ### Comment · Reviewer_Aeaf · 2024-08-09
> >
> > Thank you for addressing my concerns. I have no further questions and currently maintain my original rating.

---

> > > ### Author Response · Authors · 2024-08-10
> > > **Revising rating**
> > >
> > > We thank you for reading the rebuttal. We would request you to increase your score if you think our responses are satisfactory. If not, we would love an opportunity to discuss your concerns.

---

### Official Review · Reviewer_Xf1e · 2024-07-11

**Soundness:** 3
**Presentation:** 3
**Contribution:** 1
**Rating:** 4
**Confidence:** 5

**Summary:**

The work benchmarks the traditional methods and deep learning-based methods for medical image registration and gives a general recipe to choose registration methods.

**Strengths:**

Comprehensive experiments: The authors implement several classical variational and deep learning-based registration models on four public datasets of brain CT/MR images.

**Weaknesses:**

- The training set of deep learning models may not be sufficient because the number of image pairs is obviously not enough, which potentially makes the comparison with classical models unfair.
- Medical image registration contains a lot of content, it cannot and should not be simply represented by monomodal registration on four brain image datasets. The authors need to implement more experiments on images of more organs and tissues with monomodal and multimodal registration tasks.
- The subtitle “DLIR methods do not generalize across datasets” is almost common sense that networks trained on specific datasets cannot easily generalize to other datasets.

**Questions:**

- Please explicitly clarify the advantages compared to the well-known lean2reg challenge.
- Does the conclusion still hold on large datasets? AMOS dataset provides lots of CT and MR images can be used for model training.
- Why not submit this work to the dataset and benchmark track?

**Limitations:**

yes

---

> ### Author Rebuttal · Authors · 2024-08-06
>
> We thank the reviewer for their insightful and helpful feedback, which has significantly contributed to improving the quality of our paper. We believe we have addressed all the remaining concerns and hope the reviewer will consider increasing their score and advocating for the acceptance of our paper.
>
> **The training set of deep learning models may not be sufficient**
>
> All methods are trained on the modified OASIS training split of 364 images, yielding approximately 132k inter-subject training pairs (364 * 363). This is a substantial number of image pairs, and in practice, most DLIR methods converge within 2-3 epochs. Furthermore, most baselines like LapIRN train on an even smaller subset of OASIS, approximately 62k training pairs (250 * 249)  (although we train on 364 images for consistency). Thus, the dataset size used is sufficient for a fair comparison with classical models.
>
> **The authors need to implement more experiments on images of more organs and tissues with monomodal and multimodal registration tasks**
>
> We have already acknowledged this in the Limitations section in our paper, which forms the basis for future research. We chose the inter-subject MRI brain registration problem due to its widespread adoption by the biomedical ML community, ensuring adequate reproducibility and unbiased evaluation. While we recognize the importance of exploring other organs and tissues, none of our analysis, baselines, or evaluation protocols use domain-specific knowledge, suggesting that our results are not strictly limited to brain MRI. Expanding to more variations is a priority for future work.
>
> **The subtitle “DLIR methods do not generalize across datasets” is almost common sense**
>
> We agree, this is almost trivial in hindsight! However, seeing the results in Sec.4. and Sec.5., one might expect the domain-shift performance to fall between the results of in-distribution unsupervised and supervised training, with supervised DLIR methods outperforming classical methods to some extent. However, this is not the case (please see common rebuttal for more details), showing that large amounts of labeled data would not necessarily help on datasets with even the same modality and resolution but slightly different intensity statistics. This raises a question for large-scale, annotated data collection efforts, and addressing current limitations of DLIR methods to be robust to domain shift.
>
> **Please explicitly clarify the advantages compared to the well-known lean2reg challenge.**
>
> Learn2reg challenge which focuses on methods to maximize registration performance (measured by label overlap, landmark distance, determinant of Jacobian) on a particular dataset with data/modality specific challenges. In contrast, we perform a meta-analysis and unbiased evaluation on the factors that lead to better performance of aforementioned registration algorithms. We find that DLIR methods attribute the performance improvement to architecture, loss functions, and training specifications. However, we show that most of the improvement is attributed to being able to learn from supervised label map objectives during training. Moreover, we show that training with label maps (which are intrinsic to modality and therefore invariant to the downstream modality – see more in common rebuttal) does not necessarily imply robustness in performance to domain shift. We have modified the “Contributions” subsection to emphasize these distinctions.
>
> **Does the conclusion still hold on large datasets? AMOS dataset provides lots of CT and MR images can be used for model training**
>
> The AMOS dataset, with 500 CT and 100 MR images, is comparable to the OASIS dataset in size (414 images). We selected the OASIS dataset for reproducibility and fairness considerations, as numerous DLIR methods provide pretrained models and training recipes for OASIS. The active community around OASIS and its long-standing presence has led to detailed analyses of dataset-specific challenges (for example, registration of smaller subcortical structures). Although the AMOS dataset is relatively new, it holds significant potential for future work, and we plan to explore it in subsequent studies.
>
> **Why not submit this work to the dataset and benchmark track?**
>
> Our work goes beyond merely reproducing or benchmarking existing methods. We provide a theoretical and empirical analysis of the factors responsible for improved performance in registration tasks, particularly the access to label maps during training, and their implications for robustness across similar datasets. This paper is a meta-analysis and unbiased evaluation, similar in spirit to [7].

---

> > ### Comment · Reviewer_Xf1e · 2024-08-12
> >
> > Thanks for the response. The claimed new advantages compared to learn2reg are not convincing to me because these conclusions can also be derived from participants' algorithms. Accordingly, I slightly raise my score.

---

### Official Review · Reviewer_23gn · 2024-07-12

**Soundness:** 3
**Presentation:** 2
**Contribution:** 2
**Rating:** 4
**Confidence:** 4

**Summary:**

This paper discussed about an explicit correspondence between mutual information of the distribution and performance of classification registration methods. The authors argued that this correlation will not be affected by the learning-based methods. They validated this hypothesis on both classical and learning-based registration models and found that the weakly supervised learning-based methods performed high-fidelity intensity and label registration. Besides, they showed that these high-fidelity feature learning methods did not translate to invariance to domain shift, ending with proposing a general approach to select the best paradigm for a registration task.

**Strengths:**

**S1.** The authors experimented on the feature learning approach and their translation to invariancy to domain shift, which is an important aspect in deformable image registration. Also, they covered both classical and learning-based image registration to validate their hypothesis.

**S2.** The authors considered out-of-distribution datasets for testing and evaluating performances, which showed promising results regarding generalization.

**Weaknesses:**

**W1. Experimental findings.**

1. I appreciate the authors' finding on getting improved performance for classical methods over learning-based methods. However, I would argue that this is typically not always true considering the reported findings in the learning-based papers, e.g., VoxelMorph (VM), TransMorph (TM), etc. This registration results highly depending on the image pairs that are being considered. For example, image pairs with large age variations won't perform well in the learning-based registration tasks (please check VM, Deepflash, and related papers where they restricted the subject ages to deal with large deformations). Besides, inter-patient registration has a higher probability of getting similar results compared to the atlas-based (or pre-selected template) registration, thoroughly discussed in VM and TM papers. I found this important information missing from their experimental evaluation to support their first hypothesis (Sec. 4). I would encourage the authors to consider experimenting and reporting performances for within-patient, inter-patient, and atlas-to-patient registration results. Also, comparing the anatomical dice scores further justifies the registration models as in the existing SOTA it has been discussed about getting larger variations for critical anatomical structures such as ventricles, pallidum, cerebellum, etc.

2. I found some of the mode's performance very inconsistent with the reported version in the original papers. For example - LKU-Net is achieving more than $~0.925\pm0.025$ dice score where the authors' reported best within subject accuracy is $0.8861\pm0.01$ with an average dice score of $0.7758\pm0.0390$. The same goes for the LapIRN model's performance. There's a large difference between the reported dice in this paper vs the original paper's dice. I understand there might be different biases that might be there in terms of implementation. However, the current reported performance of these models raises questions about the credibility of the adapted implementations, preparation of the pairwise images, hyperparameter setup, etc. I would suggest the authors kindly shed some light on this part.

3. I found the supporting experiments to validate the hypothesis presented in Sec. 6 missing some important information. Did the authors perform an affine transformation on all the datasets and what are the pre-processing steps? Are they considering all 3D volumes for their experiments?


**W2. Missing large deformation diffeomorphic and related baselines.** I appreciate the authors for trying to carry out thorough experiments on different classical and learning-based registrations. However, I believe the current hypothesis would be more understandable and justifiable if the authors could initiate some experiments considering large deformation diffeomorphic-based registration methods such as LDDMM [1,2,3], where this kind of method is structured upon time-varying velocity fields, that is proven to be more robust in various deformation-based image analysis tasks.


**W3. Overclaimed hypothesis.** Statements in L332-L335 seem overstated. The authors need to discuss and experiment on within-subject/class, inter-subject/class, and atlas/template-based registration to validate the stated hypothesis in that line, which seems to be missing in the current version. Besides, the selection of the DLIR/classical registration method depends on the image analysis tasks that are being performed over image registration. Without verifying the studied registration on different image analysis tasks, it is inappropriate to come to a conclusion stated in L332.


**W4. (Minor) Technical writing.** The authors tried to aggregate all their potential findings in a structured way. But reading the whole paper kind of messed me up in understanding what are the actual contribution of this paper compared to the other survey/review papers in this domain other than performing experiments on OOD data. Overall, the presentation is kind of above the borderline but I guess if the authors tried to focus on their storyline and make their findings more clearer that would be great for the readers. For example, I found implementation details in most sections which is kind of redundant.



Overall, I appreciate the authors for working on this paper which is very relevant as well as important in the medical imaging domain, specifically in medical image registration. However, the current version of the manuscript lacks some important experimental justification and further experiments. With that being said, the current version of the manuscript is under the threshold of acceptance. However, I am open to reconsidering the initial rating if the above concerns are adequately justified.


References
----------------
[1] Yang, Xiao, Roland Kwitt, and Marc Niethammer. "Fast predictive image registration." Deep Learning and Data Labeling for Medical Applications: First International Workshop, LABELS 2016, and Second International Workshop, DLMIA 2016, Held in Conjunction with MICCAI 2016, Athens, Greece, October 21, 2016, Proceedings 1. Springer International Publishing, 2016.

[2] Shen, Zhengyang, François-Xavier Vialard, and Marc Niethammer. "Region-specific diffeomorphic metric mapping." Advances in Neural Information Processing Systems 32 (2019).

[3] Niethammer, Marc, Roland Kwitt, and Francois-Xavier Vialard. "Metric learning for image registration." Proceedings of the IEEE/CVF Conference on Computer Vision and Pattern Recognition. 2019.

[4] Wang, Jian, and Miaomiao Zhang. "Deepflash: An efficient network for learning-based medical image registration." Proceedings of the IEEE/CVF conference on computer vision and pattern recognition. 2020.

**Questions:**

Please see the weaknesses section. I tried to summarize all the findings, concerns, and questions there.

**Limitations:**

Limitations have been discussed in the paper.

---

> ### Author Rebuttal · Authors · 2024-08-06
>
> We thank the reviewer for their highly detailed and insightful feedback, this has immensely helped in improving the quality of our work. We believe we have addressed all major concerns and incorporated changes in the paper.
>
> **“I would argue that this is typically not always true considering the reported findings in the learning-based papers, e.g., VoxelMorph (VM), TransMorph (TM), etc”**
>
> We appreciate this perspective and have addressed this concern in Sec4 (Instrumentation Bias) and Fig3. VM reports a Dice score of 0.749 for ANTs using suboptimal parameters ($\sigma$ of 9px followed by 0.4px), while we observe a Dice score of 0.787 with default parameters, highlighting the need for unbiased evaluation. Our evaluation of VM uses their pretrained model and scripts, yielding consistent results w/ their paper. Other studies on lung [5] and histology [6] registration also show that classical DIR methods far outperform DLIR methods. An unbiased and unified evaluation of algorithms is necessary to fairly compare the performance gaps of said algorithms.
> Our paper finds that TM is the only DL method beating classical methods (p<0.01) on the OASIS dataset with and without Dice loss (Fig 2 and 4). TM also demonstrates good generalization among DLIR methods competing with SynthMorph, although still underperforming compared to classical methods. We urge the reviewer to review these figures and results. We are happy to clarify any remaining discrepancies in the discussion.
>
> **“Performance is different for inter-subject and atlas-based registration, image pairs with large age variations won't perform well in the learning-based registration tasks”**
>
> In general, a vast majority of methods focus on inter-subject registration, and atlas-building and atlas-based registration are considered separate problems altogether. Therefore, we keep inter-subject registration datasets a consistent theme in the paper to corroborate existing findings and avoid mixing up results from different problem statements.
>
> Re large age variations: A good registration model in general should perform well across large variations in age, subject demographics, and acquisition configurations. Classical methods performing well for age variations further strengthen our findings in Sec.4 and Sec.6, i.e. labeled data becomes necessary in these settings for DLIR methods to perform better. We have added these observations into the paper further motivating our study.
>
> **“consider experimenting and reporting performances for within-patient, inter-patient, and atlas-to-patient registration results”**
>
> These are interesting experiments indeed and form the scope for future work, along with multimodal registration, and registration of different anatomy like chest CT or abdomen. Due to space constraints, fairness and reproducibility considerations, we choose OASIS dataset for training since a large number of SOTA DLIR methods provide training recipes for this dataset, and IBSR/CUMC/MGH/LPBA datasets that have been extensively used in unbiased community-standard neuroimaging evaluations [7,9].
>
> **“model performance is very inconsistent with the reported version in the original papers, LKU-Net is achieving 0.77 average Dice… same with LapIRN model's performance”**
>
> We’re not sure where these numbers come from. In Fig.3 we show that the LKU paper reports a 0.88 Dice score, and our evaluation shows a 0.904 Dice score – Fig.4. (top) shows that LKU-Net is the top performer with Dice supervision. Similarly, we show in Fig2. that LapIRN reports 0.808 Dice but we report 0.788 – a reasonable deviation. In fact, LapIRN performance improves significantly when Dice supervision is added (Fig2 v/s Fig4), the latter of which the original paper did not implement or report.
>
> Our work takes great care to accurately reproduce and not misrepresent the performance of DLIR and classical methods. We ask the reviewer to review the subsection on Instrumentation bias (Sec 4) – we’re confident they will find DLIR methods represented accurately. DLIR methods may misrepresent classical baselines by choosing suboptimal hyperparameters; we work towards a more fair, unbiased and unified evaluation of classical and DLIR methods under the same umbrella.
>
> **“hypothesis presented in Sec 6 missing some important information”**
>
> For all datasets, we follow the preprocessing steps followed by Klein et.al. [7]. We have added this detail in the paper.
>
> **“W2. Missing large deformation diffeomorphic and related baselines”**
>
> Classical methods like ANTs, Greedy, FireANTs build on the LDDMM framework to avoid explicitly storing the full 4D velocity fields by integrating the infinitesimal velocities using gradient descent, & are shown to model *very* large deformations by independent evaluations [7,8]. Moreover, these are widely used classical baselines used by the broader biomedical community, and should not be seen as a reason to invalidate the results in the paper.
>
> **“W3. Overclaimed hypothesis”**
>
> We have changed L332-335 to reflect the conclusions for inter-subject registration, which is a defacto standard for registration. Since none of the analysis, baselines, and evaluation make any domain/subject-specific modeling assumptions, and the datasets are community-standard benchmarks, the results are valuable, both within the neuroimaging and the biomedical communities at large.
>
> **“W4. (Minor) Technical writing”**
>
> The contribution of the paper is to recalibrate claims about DLIR methods' performance by isolating label supervision as the primary contributor to their superiority. This motivates the question if label supervision can be robust to domain shift and finds it is not, prompting a reconsideration of data collection pipelines to improve registration performance, and rethinking DLIR design decisions to be robust to domain shift. We have made these changes in the contributions and discussion sections.

---

> > ### Comment · Reviewer_23gn · 2024-08-11
> > **Official Comment by Reviewer 23gn**
> >
> > I thank the reviewers for their response. After reading the rebuttal, I have the following standings —
> >
> > - Some of my concerns regarding LDDMM and the performances of different existing models have been adequately addressed.
> > - The paper's contribution is limited considering the hypothesis that the authors evaluated in this paper.
> > - I agree with my other reviewer that this paper is more aligned with the Dataset and Benchmarking track. Besides, I agree that experiments on datasets other than Brain could reinforce the contribution of their standings. More experiments related to intra-subject, atlas-based registration could further rectify their claims.
> >
> > *After reading all the reviewers' comments and the rebuttal, I think the paper is still on the Borderline (keeping my score as it is), considering the technical contribution, carried out experimentations, overall writing, and the submission track.*
> >
> > I suggest that the authors to address the findings from all reviewers in their revised version.

---

### Official Review · Reviewer_tuv2 · 2024-07-13

**Soundness:** 4
**Presentation:** 3
**Contribution:** 3
**Rating:** 6
**Confidence:** 3

**Summary:**

The manuscript investigates the characteristics of two types of registration approach based on traditional variational optimization and deep learning. Experiments revealed a correlation between the mutual information of the distribution of per-pixel intensity and labels, and the performance of classical registration methods. Then the manuscript argues unsupervised deep learning does not improve label matching performance compared to traditional methods, whereas supervised learning methods show improved label matching. Lastly they show learning methods do not generalize well.

**Strengths:**

1. This study reflects original thinking, trying to draw insights on a challenging topic.
2. Experiments design is well motivated to study the hypothesis.
3. The results challenge a claim made by a number of existing studies "learning methods can provide improve label matching when optimized in an unsupervised fashion"

**Weaknesses:**

1. Two of the three claims discussed herein seem trivial. It is not surprised that "Supervised DLIR methods demonstrate enhanced label matching" and "DLIR methods do not generalize across datasets".
2. I feel that the abstract and intro set up a high expectation by saying "we propose a general recipe to choose the best paradigm for a given registration problem, based on these observations." but then again it is a one-sentence recipe in the end that is not surprising to readers "a practitioner should choose DLIR methods only if they have access to a large labeled dataset, and their application is limited to the same dataset distribution. In all other cases, classical optimization-based methods are the more accurate and reliable choice"

**Questions:**

In Fig. 1, shouldn't there also be a correlation within each individual dataset?

**Limitations:**

While the intro and abstract suggest a study on general registration problems, all experiments are based on brain MRI. It is not clear whether the problem/hypothesis/conclusion is specific to brain MRI.

---

> ### Author Rebuttal · Authors · 2024-08-06
>
> We thank the reviewer for their time and insightful feedback, and are pleased to note the recognition of the originality of our work, the well-motivated design of our experiments, and the challenge posed to existing claims in the DLIR literature. We address some of their concerns below, and hope that their review and discussion can help convince all reviewers of the importance of our work, and champion our paper.
>
> **Two of the three claims seem trivial.**
>
> We agree that the claims seem straightforward in hindsight. However, we contextualize this conclusion from the claims made by existing DLIR literature, i.e. a carefully selected intensity loss function, network architecture, pretraining schema and large datasets lead to SOTA performance on alignment of anatomical regions. A variety of studies cited in our paper report results with and without label supervision, which we show is the confounding variable that leads to a significant improvement in performance. Therefore, performance of DLIR methods with and without Dice score must be compared to justify the factors involved in improving performance of DLIR methods.
> Our analysis in Section 3 elucidates that unsupervised methods do not benefit from advancements in network architecture, a point further verified in Section 4.
> Section 5 is included to underscore the fact that DLIR methods’ superior performance indeed comes from incorporating label supervision during training, a capability absent in classical methods.
>
> Section 6 seems trivial in the context of our paper, but for the biomedical community at large, pretrained models and large datasets are built to train models that generalize outside the training domain (see common rebuttal). Our findings in Section 6 suggest a need to reassess huge labeled data collection efforts and to formulate DLIR advancements that exhibit robustness against minor domain shifts instead. This reevaluation is crucial for leveraging large amounts of annotated data effectively. We have added this both in the Motivation and Discussion sections.
>
> **“I feel that the abstract and intro set up a high expectation..”**
>
> We acknowledge that the phrasing of our contribution may have set an overly ambitious expectation. We have revised the statement in the paper to: “We reassess and recalibrate performance expectations from classical and DLIR methods under access to label supervision and training time, and its generalization capabilities under minor domain shifts.” This modification more accurately reflects the scope and implications of our study.
>
> **“shouldn't there also be a correlation within each individual dataset”**
>
> We find that since the labeled regions are fixed within a dataset, the variation between MI and registration score is dominated by other intra-dataset factors. The correlation between MI and registration performance occurs at the granularity of the dataset, which is shown in Fig.3. This granularity highlights the broader applicability of our findings across different datasets rather than within a single dataset.
>
> **“are the hypothesis/conclusion specific to brain MRI”**:
>
> We assert that the methods and evaluation metrics employed in our study do not incorporate domain-specific information, making our observations broadly applicable. Brain MRI was chosen due to the extensive availability of solutions and open-access resources provided by the neuroimaging and machine learning communities. This choice ensures that our analysis is unhindered by fairness or reproducibility concerns.

---

> > ### Comment · Reviewer_tuv2 · 2024-08-11
> >
> > Thanks for clarification. I'm still positive about the paper after reading all responses and therefore remain my rating.

---

### Author Rebuttal · Authors · 2024-08-06

We thank all reviewers for their insightful feedback and for taking time to improve the quality of our work. We are glad that reviewers found our work reflecting original thinking and drawing useful insights [tuv2], well motivated problem [Aeaf], consideration of both classical and DLIR methods [23gn], well motivated and well designed experiments [tuv2, Xf1e, Aeaf], clear writing [Aeaf]. We have addressed all questions in the individual comments. We summarize and clarify some common concerns:

**Some of the claims are trivial**

We agree that these claims look trivial in hindsight. However, these claims must be contextualized from the perspective of claims made by existing DLIR literature. Specifically, most DLIR methods propose either a new architecture, training recipe (multi-scale, cascaded, or pre-training), or loss functions, and claim that these methodologies lead to superior performance over classical registration methods. However, the argument in Section 3 / Fig2. indicates that these contributions do not add any information on the existing correlation between mutual information and dice score. Hence, the primary driver of performance improvement is anatomical label supervision.

Our work provides a unified and fair evaluation of classical and DLIR methods without label supervision (Sec 4), confirming that the conclusions from Section 3/Figure 2 hold true. An ablation study over label supervision (Sec 5) further verifies that label supervision is indeed the confounding variable leading to superior performance. Thus, Sections 4 and 5 are pivotal in isolating the real confounding variable, both theoretically and empirically.

Importantly, anatomical labels are intrinsic to the anatomy and not inherently modality-dependent, making them domain-invariant. This motivation for using synthetic data with consistent label maps is often used to train domain-invariant models [1][2]. Our key non-trivial finding, presented in Section 6, reveals that while DLIR models trained with label supervision may perform worse under domain shift (DS) compared to in-distribution (ID) scenarios, the expected performance hierarchy does not hold:

Expected: (DLIR Sup ID) > (DLIR Sup DS) > (Classical)

Observed: (DLIR Sup ID) > (Classical) > (DLIR Sup DS)

This significant finding highlights that DLIR methods do not maintain their performance advantage over classical methods under domain shift. This result has profound implications for annotated data collection and challenges the notion that large labeled datasets ensure robust generalization. For example, Learn2Reg 2023 [3] mentions “By providing easy-to-use solutions applicable to a variety of medical registration problems, we hope to strengthen both generalization and comparability between them.”. However, our findings underscore that the generalization of DLIR methods to domain shifts is heavily understudied and generally negative. This necessitates a reassessment of expensive labeled data collection efforts and emphasizes the need for developing DLIR methods, like those proposed in [1], that are more robust to domain shifts.
We have added a subsection in Section 3 summarizing these motivations and the broader implications of our findings. We hope this better contextualizes and emphasizes the significance of our results.

**Are the findings applicable to brain MRI only?**

We assert that the methods and evaluation metrics employed in our study do not incorporate domain-specific information, making our observations broadly applicable. Brain MRI was chosen due to the extensive availability of solutions and open-access resources provided by the neuroimaging and machine learning communities, also somewhat inspired by [7]. This choice ensures that our analysis is unhindered by fairness or reproducibility concerns.

**What about other modalities / anatomy / intra-subject / atlas-based settings?**

We have already acknowledged this in the Limitations section in our paper, forming the basis for future research. We chose the inter-subject MRI brain registration setup due to its widespread adoption by the biomedical ML community, ensuring adequate reproducibility with unbiased and fair evaluation. Registration literature in other anatomy and modalities – hippocampus - Table 1 in [4], lung images [5], and histology [6] also seem to suggest similar results (classical methods outperform DLIR in unsupervised settings), but a more comprehensive study with unified training and inference setup is necessary and forms the motivation for future work.

[1] Hoffmann, Malte, et al. "SynthMorph: learning contrast-invariant registration without acquired images." IEEE transactions on medical imaging 41.3 (2021): 543-558.

[2] Dey, Neel, et al. "AnyStar: Domain randomized universal star-convex 3D instance segmentation." Proceedings of the IEEE/CVF Winter Conference on Applications of Computer Vision. 2024.

[3]  https://learn2reg.grand-challenge.org/learn2reg-2023/

[4] W. Zhu, Y. Huang, D. Xu, Z. Qian, W. Fan and X. Xie, "Test-Time Training for Deformable Multi-Scale Image Registration," 2021 IEEE International Conference on Robotics and Automation (ICRA)

[5] Fu, Yabo, et al. "LungRegNet: an unsupervised deformable image registration method for 4D‐CT lung." Medical physics 47.4 (2020): 1763-1774.

[6] Borovec, Jiří, et al. "ANHIR: automatic non-rigid histological image registration challenge." IEEE transactions on medical imaging 39.10 (2020): 3042-3052.

[7] A. Klein, et al. Evaluation of nonlinear deformation algorithms applied to human brain MRI registration. NeuroImage, 46(3):786–802, July 2009.

[8] Murphy, Keelin, et al. "Evaluation of registration methods on thoracic CT: the EMPIRE10 challenge." IEEE transactions on medical imaging 30.11 (2011): 1901-1920.

[9] Mok, Tony CW, and Albert Chung. "Affine medical image registration with coarse-to-fine vision transformer." Proceedings of the IEEE/CVF Conference on Computer Vision and Pattern Recognition. 2022.

---

### Decision · Program_Chairs · 2024-09-25

**Decision:**

Accept (poster)

**Comment:**

This work investigates the classical optimisation and learning-based methods in deformable image registration, with the aim to gain more understanding and insights into the performance, generalisation, and the working conditions of the two typical approaches to this task. Reviewers recognise that this work has merits of original thinking, well motivated experimental design, results that challenge existing understanding, considering out-of-distribution test case, and rich experiments making people rethink. At the same time, the reviewers raise the following issues, including 1) some claims that seem trivial; 2) the need to consider more registration settings; 3) the inconsistency of performance reported; 4) missing baselines and overclaimed hypothesis; 5) the insufficiency of the training set; 6) more experiments on organs, tissues, and modalities; 7) missing new solutions or deeper insights; and 8) not pointing out the potential future direction. The authors have made efforts to provide response to each of the raised issues. Among the four reviewers, after reading the rebuttal, one reviewer is still positive (with rating 6); one reviewer thinks this work is still on the borderline (with rating 4); one reviewer is not fully convinced and slightly raises the rating to 4; and the last reviewer has no further questions and maintains the original rating of 5.

By reading the reviews, rebuttal and the submission, AC thinks that the rebuttal is overall clear, sound and effective. Furthermore, as indicated by two reviewers, this work reflects original thinking and makes people rethink the development of learning-based registration methods. Although some claims do not seem to be surprising enough, validating them via systematic investigation has its value and significance to the relevant research community. This kind of research shall be encouraged in order to better understand the many existing methods. Considering all the factors, AC recommends this submission for acceptance. Meanwhile, the authors shall adequately incorporate the key points in the rebuttal to further strengthen this work, including the presentation.